# A Systematic Review of the Role of Prebiotics and Probiotics in Autism Spectrum Disorders

**DOI:** 10.3390/medicina55050129

**Published:** 2019-05-10

**Authors:** Qin Xiang Ng, Wayren Loke, Nandini Venkatanarayanan, Donovan Yutong Lim, Alex Yu Sen Soh, Wee Song Yeo

**Affiliations:** 1MOH Holdings Pte Ltd., 1 Maritime Square, Singapore 099253, Singapore; wayren.loke@mohh.com.sg (W.L.); nandini9293@icloud.com (N.V.); 2KK Women’s and Children’s Hospital, 100 Bukit Timah Rd, Singapore 229899, Singapore; 3National University Hospital, National University Health System, Singapore 119074, Singapore; alex_ys_soh@nuhs.edu.sg (A.Y.S.S.); paeyws@nus.edu.sg (W.S.Y.); 4Department of Child and Adolescent Psychiatry, Institute of Mental Health, 10 Buangkok View, Singapore 539747, Singapore; Donovan_LIM@imh.com.sg; 5Yong Loo Lin School of Medicine, National University of Singapore, Singapore 117597, Singapore

**Keywords:** probiotics, *Lactobacillus*, *Bifidobacterium*, autism, ASD, gut microbiota

## Abstract

*Background*: Autism spectrum disorder (ASD) is a complex developmental condition typically characterized by deficits in social and communicative behaviors as well as repetitive patterns of behaviors. Despite its prevalence (affecting 0.1% to 1.8% of the global population), the pathogenesis of ASD remains incompletely understood. Patients with ASD are reported to have more frequent gastrointestinal (GI) complaints. There is some anecdotal evidence that probiotics are able to alleviate GI symptoms as well as improve behavioral issues in children with ASD. However, systematic reviews of the effect of prebiotics/probiotics on ASD and its associated symptoms are lacking. *Methods*: Using the keywords (prebiotics OR probiotics OR microbiota OR gut) AND (autism OR social OR ASD), a systematic literature search was conducted on PubMed, EMBASE, Medline, Clinicaltrials.gov and Google Scholar databases. The inclusion criteria were original clinical trials, published in English between the period 1st January 1988 and 1st February 2019. *Results*: A total of eight clinical trials were systematically reviewed. Two clinical trials examined the use of prebiotic and/or diet exclusion while six involved the use of probiotic supplementation in children with ASD. Most of these were prospective, open-label studies. Prebiotics only improved certain GI symptoms; however, when combined with an exclusion diet (gluten and casein free) showed a significant reduction in anti-sociability scores. As for probiotics, there is limited evidence to support the role of probiotics in alleviating the GI or behavioral symptoms in children with ASD. The two available double-blind, randomized, placebo-controlled trials found no significant difference in GI symptoms and behavior. *Conclusion*: Despite promising preclinical findings, prebiotics and probiotics have demonstrated an overall limited efficacy in the management of GI or behavioral symptoms in children with ASD. In addition, there was no standardized probiotics regimen, with multiple different strains and concentrations of probiotics, and variable duration of treatments.

## 1. Introduction

Autism spectrum disorder (ASD) is a complex developmental condition typically characterized by deficits in social and communicative behaviors as well as repetitive patterns of behaviors [1]. Worldwide, prevalence estimates for ASD range from 0.1% to 1.8% [2]. In the United States, the estimated prevalence of ASD is 16.8 per 1000 (one in 59) children aged eight years old [3]. The etiology behind ASD is not known for certain, but current studies suggest that there is an intricate interplay of several genetic, epigenetic and environmental factors [4,5]. Patients with ASD also develop multiple comorbidities ranging from psychiatric issues such as anxiety [6] to the commonly seen gastrointestinal (GI) disorders [7]. There is an increased incidence of GI complaints in ASD patients [8] and the symptoms are variable, presenting as diarrhea, constipation, abdominal pain or bloating. These GI disorders can be rather refractory to conventional treatment [9] and therefore present a challenge to treat.

Patients with ASD have been reported as having different compositions of gut microbiota compared to neurotypical controls. There have been observations of higher concentrations of pathogenic *Clostridium* bacteria [10], a decreased *Bacteroides/Firmicutes* ratio, and increased *Lactobacillus* and *Desulfovibrio* species [11,12]. The effect of gut microbiota on the GI system has been well documented, influencing GI motility, intestinal epithelium permeability and mucus production [13]. Specifically, in patients with ASD, the severity of GI symptoms have been linked with derangements in the gut microbiota, such as during administration of antibiotics. It was also noted that the GI and behavioral symptoms reverted once antibiotics were stopped [14]. This opens up further avenues of research for the role of gut microbiota-altering agents such as probiotics as a potential therapeutic option.

As defined by the International Scientific Association for Probiotics and Prebiotics (ISAPP), probiotics are “live microorganisms that, when administered in adequate amounts, confer a health benefit on the host” [15]. Probiotics aim to restore normal balance of human gut microbiota and have been shown to be effective in treating other GI disorders such as traveler’s diarrhea [16] and irritable bowel syndrome [17]. Recent studies have suggested that probiotics are beneficial in treating several psychological symptoms such as depression and anxiety [18,19]. It is theorized that there exists a complex interplay between the brain and the GI tract termed the “gut–brain axis”. Gut microbiota play an important role in modulating this gut–brain axis and dysbiosis can have negative effects on not only the GI tract, but also psychological symptoms [20]. Dietary exclusions and supplements have been investigated in the management of ASD symptoms [21]. In particular, there is some anecdotal evidence that probiotics are able to alleviate GI symptoms as well as improve behavioral issues in children with ASD [22].

Besides probiotics, the impact of prebiotics on the gut microbiota is not to be neglected. As defined by the ISAPP, prebiotics are substrates “selectively utilized by host microorganisms, conferring a health benefit” [15]. Examples are non-digestible carbohydrates. The effect of exclusion diets and prebiotics was recently evaluated in children with ASD, with findings including significant changes in gut microbiota composition and metabolism and amelioration of GI and behavioral symptoms [23].

Despite significant public interest in pre- and probiotics, systematic reviews of the effect of prebiotics/probiotics on ASD and its associated behavioral/GI symptoms are lacking. Therefore, this review aims to examine the clinical role of prebiotics and probiotics in the management of GI and core ASD symptoms.

## 2. Methods

A systematic literature search was performed in accordance with Preferred Reporting Items for Systematic Reviews and Meta-Analyses (PRISMA) guidelines. Using the keywords (prebiotics OR probiotics OR microbiota OR gut) AND (autism OR social OR ASD), a preliminary search on the PubMed, EMBASE, Medline, Clinicaltrials.gov and Google Scholar databases yielded 1592 papers published in English between 1st January 1988 and 1st September 2018. Attempts to search grey literature were made using the Google search engine. Title/abstract screening were performed independently by the researchers (Q.X. Ng, W.R. Loke and N. Venkatanarayanan) to identify articles of interest. For relevant abstracts, full articles were obtained, reviewed and also checked for references of interest. If necessary, the authors of the articles were contacted to provide additional data.

The inclusion criteria for this review were: (1) published retrospective or prospective study (excluding single case reports), (2) patients with clinical diagnosis of ASD, (3) trial of prebiotic and/or probiotic treatment, and (4) available data on GI/behavioral/psychiatric symptoms. Data such as study design, study population and demographics, and outcome measure were extracted. The primary outcome measure of interest was GI/behavioral/psychiatric symptom outcome with pre-/probiotic treatment.

## 3. Results

The abstraction process is summarized in Figure 1.

A total of eight clinical trials were systematically reviewed (Table 1). Two clinical trials examined the use of prebiotic and/or diet exclusion while six involved the use of probiotic supplementation in children with ASD. Most of these were prospective, open-label studies, with the exception of three randomized, double-blind, placebo-controlled trials.

## 4. Discussion

Overall, there is a general paucity of randomized, controlled trials examining the use of prebiotics or probiotics in children with ASD. Only two clinical trials examined the use of prebiotics in children with ASD [23,24]. Beta-galacto-oligosaccharide prebiotic (B-GOS) did not have a significant effect on GI symptoms, while bovine colostrum product improved certain GI symptoms and produced significant improvements in irritability scores and stereotypy. When B-GOS was combined with an exclusion diet (gluten and casein free), the results showed a significant reduction in anti-sociability scores [23].

With regard to probiotic supplementation, based on a systematic review of current evidence, there is overall limited evidence to support the efficacy of probiotics in alleviating the gastrointestinal or behavioral symptoms prevalent in children with ASD. Only two open-label trials found significant improvement in gastrointestinal symptoms after probiotic supplementation [27,29]. Three open-label trials found that probiotics also help to mitigate the behavioral issues characteristic of ASD [25,27,29]. However, the two available double-blind, randomized, placebo-controlled trials found no significant difference in GI symptoms [26] and behavior [26,28]. The clinical utility of prebiotics and probiotics in management of GI or behavioral symptoms in patients with ASD remains to be validated by future clinical studies.

The exact mechanism by which probiotics exert potential therapeutic effects is not known but there are several hypotheses. The most direct answer is that patients with ASD possess significantly altered gut microbiota [10,11,12]; it is this altered gut microbiota that stems the gastrointestinal issues. Probiotics would help to restore gut microbiota back to normal levels, in the process ameliorating concomitant gastrointestinal symptoms. Probiotics have been shown to prevent *Candida* colonization in the gut [30] and one of the studies found reduced *D*-arabinitol levels, a metabolite of *Candida* species, in the urine of ASD children after probiotic supplementation [25]. Another one of the studies reviewed reported a reduction of *Clostridium* species in the stool samples of children who received probiotics [27].

Probiotics presumably reduce gut inflammation through a variety of mechanisms, such as reducing gut barrier permeability, downregulating inflammatory cytokines and other immunomodulatory effects. Raised levels of mucosal inflammation has been linked to several gastrointestinal disorders, including irritable bowel syndrome and inflammatory bowel disease [31,32]. Children with ASD experience gastrointestinal symptoms and have been shown to have higher levels of gut immune inflammation that is associated with gut dysbiosis [32,33]. Therefore, probiotics could have a role in restoring gut microbiota as well as lowering levels of gut inflammation.

From the studies examined, there is some evidence that prebiotic and probiotic supplementation resulted in an improvement of behavior of the children with ASD [24,25,27,29]. The effect of probiotics on the brain is not new, with several studies showing its benefit in treating several psychological conditions such as depression and anxiety [18,19]. The main postulation behind this is that probiotics act via the gut–brain axis to influence neurotransmission and mood states. Probiotics are able to influence several neuroactive metabolites such as gamma-aminobutyric acid (GABA) and serotonin [18,34]. Serotonin has been shown to be influenced by gut bacteria and also specifically in individuals with ASD, there is hyperactivation of a gene that codes for serotonin reuptake transporters [34]. Studies have also linked the neuropeptide oxytocin to social behavior and the pathogenesis of ASD [35,36]. This may be how probiotics influence the behavior of ASD children. Research in this area is currently limited and most of the data were secondary objectives and not thoroughly examined. Hence, there is a need for further research on this front.

Microbiome derangements may also contribute to dysregulated immunity and lead to autoantibodies formation. *Lactobacillus* species have been found to down-regulate hypersensitive responses [37]. The putative molecular mimicry mechanism of Pediatric Autoimmune Neuropsychiatric Disorder Associated with *Streptococcal* infections (PANDAS), which presents with the sudden onset of obsessive-compulsive disorder (OCD) or tics (and sometimes co-occurs with autism), is an example of how microbial dysbiosis is thought to incite dysregulated immunity and autoantibodies formation, ultimately resulting in behavioral abnormalities [38].

For future work, mechanistic investigations employing the use of ‘multi-omics’ could be planned. Recent technological advancements in the area of metabolomics have vastly improved the sensitivity and accuracy at which metabolites can be detected and characterized [39,40]. These metabolites may be microbial by-products that could influence gene regulation, neuronal transmission, or other biochemical perturbations in individuals with autism. Previous studies have implicated microbial metabolites such as 4-ethlyphenyl sulfate (4EPS) [41,42] and propionic acid (PPA) [43]. 4EPS was found to be significantly higher in the maternal immune activation (MIA) mice offspring with autistic-like features [41], and oral supplementation with the human commensal *Bacteroides fragilis* corrected the level of 4EPS, improved gut permeability, and alleviated autistic-like behaviors, compared with controls. Injecting 4EPS into the bloodstream of healthy mice also induced anxiety-like behavior similar to that of MIA mice [41]. Rats treated with PPA, a fermentative by-product of some gut microbial species and food preservative, showed autistic-like behavior [43].

Although prebiotics and probiotics have demonstrated promising findings in preclinical and animal studies [44,45], the same cannot be said of present clinical studies. A large majority of available studies were also limited in sample size, with most of them being single-center trials enrolling only 20 to 30 children. Also, many of these studies were open-label trials and relied on qualitative, self-reported questionnaires and surveys to gauge treatment response, inviting potential bias into the studies. There might also be various difficulties experienced by the parents in evaluating these aspects, especially owing to the communication deficits typical of children with ASD. More randomized, controlled studies with a larger study population and the use of clinician scoring may lead to more robust studies and results. 

In addition, there was no standardized probiotics regimen, with multiple different strains and concentrations of probiotics, and variable duration of treatments. The gut biome is known to vary based on geographical location [46] and the efficacy of probiotics is also known to be strain- and condition-specific [47]. Further research involving a standardized intervention regimen is warranted.

With regard to the safety of probiotic administration, probiotics have been applied and used safely in food and dairy products and the available evidence does not indicate an increased risk, but there are anecdotal reports that probiotics may worsen outcomes, for example, in patients receiving radiotherapy [48]. The current literature cannot confidently state the safety of probiotics as there is a paucity of systematic reporting of adverse events [49].

## 5. Conclusions

Based on current studies, there is overall limited evidence to support the role of prebiotics or probiotics in alleviating GI or behavioral symptoms in children with ASD. However, the dearth of studies does not mean that there is no effect. Two open-label trials found significant improvement in GI symptoms after probiotic supplementation, and three open-label trials found that probiotics helped to mitigate the behavioral issues characteristic of ASD. However, the two available double-blind, randomized, placebo-controlled trials found no significant difference in GI symptoms and behavior. Despite promising preclinical findings and positive anecdotal reports, the clinical utility of prebiotics and probiotics in the management of GI or behavioral symptoms in children with ASD remains to be validated by future studies. There is a lack of randomized, controlled trials and most studies had a small sample size (*N* < 50). In addition, there was no standardized probiotics regimen, with multiple different strains and concentrations of probiotics, and variable duration of treatments. Looking ahead, further mechanistic studies enabled by the advent of metabolomics and larger trials involving a standardized intervention regimen are necessary to advance the field.

## Figures and Tables

**Figure 1 medicina-55-00129-f001:**
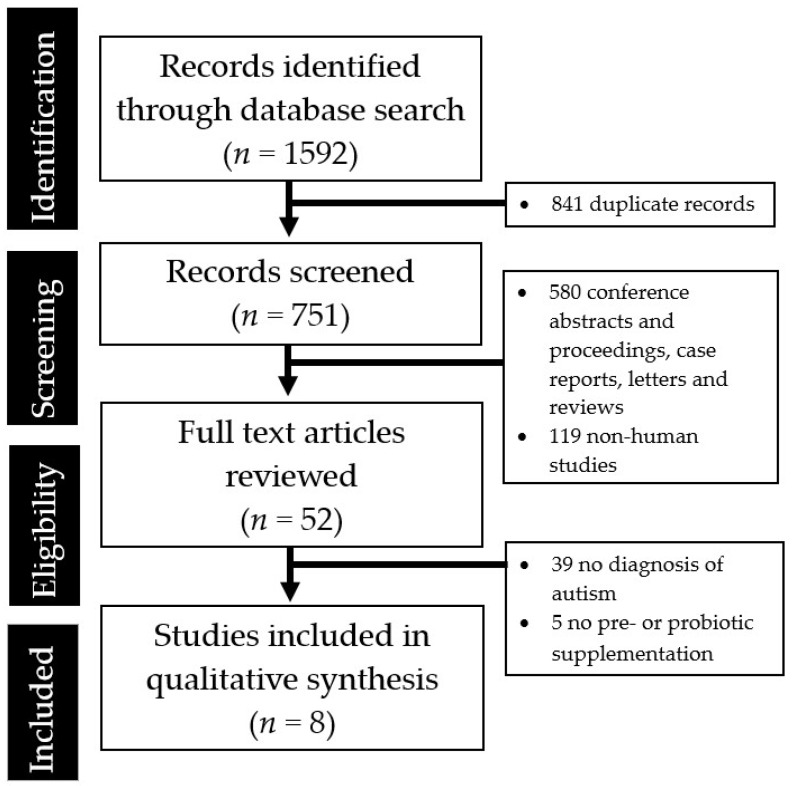
Preferred Reporting Items for Systematic Reviews and Meta-Analyses (PRISMA) flow diagram summarizing the studies identified during the literature search and abstraction process.

**Table 1 medicina-55-00129-t001:** Clinical trials involving the use of pre/pro-biotics in children with autism spectrum disorder (ASD) (arranged alphabetically by first author’s last name).

Author, Year	Study Design	Sample Size (*N*)	Study Population	Prebiotic or Probiotics Strains	Study Duration	Key Findings
**Prebiotics**						
Grimaldi 2018 [23]	Randomized, double-blind, placebo-controlled	41	4–11 year old children with ASD, 75% male, UK	Maltodextrin— GLUCIDEX^®^; 1.8 g	6 weeks	Metabolic shifts were observed in urine spectra profile and faecal samples after B-GOS^®^ intervention. Reduced gastrointestinal (GI) discomfort but no significant difference in GI symptoms or sleep (volunteer diaries).
Sanctuary 2019 [24]	Randomized, double-blind, cross over study	8	2–11 year old children with ASD, 87.5% male, US	Bovine colostrum product (BCP) *Bifidobacterium longum* supbsp. *infantis* (UCD272)	Once daily for 12 weeks	Reduced frequency of certain GI symptoms in both groups (BCP only vs BCP + *B. infantis*) Significant improvement in irritability scores and stereotypy in the group that received only BCP. Improvements may be due to a reduction in IL-13 and TNF-α production
**Probiotics**						
Kaluzna-Czaplinska 2012 [25]	Prospective, open-label	22	4–10 year old children with ASD, 90% male, Poland	*Lactobacillus acidophilus*	Twice daily for 1 month	Probiotics reduced the D-arabinitol and the ratio of D-/L-arabinitol (DA/LA) in the urine of children with autism. Significant improvement in the ability to concentrate and carry out orders.
Parracho 2010 [26]	Randomized, double-blind, placebo-controlled	39	4–16 year old children with ASD, UK	*Lactobacillus plantarum*	3 weeks	Lower levels of *Clostridium* in stools after probiotics No major differences for behaviour (DBC-P scores) No major differences in GI symptoms (volunteer diaries)
Shaaban 2018 [27]	Prospective, open-label	30	5–9 year old children with ASD, 63% male, Egypt	*Lactobacillus acidophilus, Lactobacillus rhamnosus, Bifidobacteria longum*	Once daily for 3 months	Increased levels of *Bifidobacteria* in stool samples after probiotics Improved behaviour after probiotics (ATEC scores) Improvement of GI symptoms (6-GSI scores)
Slykerman 2018 [28]	Two-center, randomized, double-blind, placebo-controlled	342	Children followed from birth to 11 years, New Zealand	*Lactobacillus rhamnosus, Bifidobacteria animalis,* *Bifidobacterium lactis HN019*	Mothers given probiotics from 35 weeks pregnant until 6 months. Children receive treatment from birth to 2 years	Worse behaviour with probiotics (BRIEF Behaviour regulation and CES-DC scores)
Tomova 2015 [12]	Prospective, open-label, controlled	29	Children with ASD from 2–9 years old, siblings of ASD children 5–17 years old, control children 2–11 years old, Slovakia	3 strains of *Lactobacillus,* 2 strains of *Bifidobacteria*, 1 strain of *Streptococcus*	3 times daily for 4 months	Reduced levels of *Bifidobacteria* and *Lactobacillus* after probiotics Reduced levels of TNF-α in stools after probiotics Increased TNF-α levels linked to increased GI symptoms and ASD severity
West 2013 [29]	Prospective, open-label	33	3–16 year old children with ASD, USA	*Lactobacillus acidophilus, Lactobacillus casei, Lactobacillus delbrueckii, Bifidobacteria longum, Bifidobacteria bifidum*	3 times daily for 21 days	Improvement in behaviour (ATEC scores and participants comments) Improvement in GI symptoms, specifically constipation and diarrhea (ATEC scores and stool diary and questionnaire)

6-GSI, 6-Item Gastrointestinal Severity Index; ATEC, Autism Treatment Evaluation Checklist Structure; B-GOS, beta-galacto-oligosaccharide prebiotic; BRIEF, Behavior Rating Inventory of Executive Function; CES-DC, Center for Epidemiological Studies Depression Scale for Children; DBC, Developmental Behaviour Checklist; IL, interleukin; TNF-α, tumor necrosis factor alpha.

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
