# Peer review of "A Systematic Review of the Role of Prebiotics and Probiotics in Autism Spectrum Disorders"

_medicina, 2019, doi:10.3390/medicina55050129_

Round 1

Reviewer 1 Report

This manuscript makes a systematic review of studies using probiotics and prebiotics to treat autism and its symptoms. It is an interesting study in a world where there are a lot of speculations, and few useful studies trying to prove a role for gut microbiota interventions and ASD. Below you can see my major comments.

Pg 1, L39: authors show the prevalence in the USA, which do not always reflect the worldwide profile of certain pathologies. Also, from the studies used in this systematic review, just one is American and there are more from other countries. It would be more informative if the authors could show data from different countries (at least the cited in this manuscript), or a summary of the prevalence in the world with data from an international health body (eg. WHO).

Pg 2, L64: authors argue about evidence of probiotic ameliorating ASD symptoms but only provide as reference a review. Authors could indicate a proper study instead.

Pg 2, L67: there is no explanation of what prebiotics is, and the differences with probiotic.

Pg 2, L71: an explanation of the importance of a systematic review in this field is missing.

Pg 2, L72: I suspect that sometimes probiotics are not called probiotics (especially when a single bacteria is used). How did the authors check for this issue?

Pg 2, L75: where the PRISMA guideline can be found?

Pg 2, L79: are there additional studies to be added since September? If yes, please update! Maybe these ones? https://www.ncbi.nlm.nih.gov/pubmed/30625189 and https://www.ncbi.nlm.nih.gov/pubmed/30743242

Pg 5, L137: the authors cited the serotonin as an important pathway linking gut-brain axis and autism. This affirmation is actually based in a review, not an actual study. Moreover, there are other links, perhaps even stronger, that could explain the role of pre/probiotics and autism (eg. oxytocin).

P6, L162: authors mentioned that there are several preclinical and clinical studies with pre-probiotic and autism but cite just one manuscript. Are there others (yes!) or should you rephrase the sentence?

P6, L190: how metabolomics can help to understand the mechanisms of pre-probiotics?

Author Response

Reviewer 1

This manuscript makes a systematic review of studies using probiotics and prebiotics to treat autism and its symptoms. It is an interesting study in a world where there are a lot of speculations, and few useful studies trying to prove a role for gut microbiota interventions and ASD. Below you can see my major comments.

Pg 1, L39: authors show the prevalence in the USA, which do not always reflect the worldwide profile of certain pathologies. Also, from the studies used in this systematic review, just one is American and there are more from other countries. It would be more informative if the authors could show data from different countries (at least the cited in this manuscript), or a summary of the prevalence in the world with data from an international health body (eg. WHO).

REPLY: Thank you for the comment. We have now added the sentence, “Worldwide, prevalence estimates for ASD range from 0.1% to 1.8% [2].”

Pg 2, L64: authors argue about evidence of probiotic ameliorating ASD symptoms but only provide as reference a review. Authors could indicate a proper study instead.

REPLY: Thank you for the comment. A proper study is now cited.

Pg 2, L67: there is no explanation of what prebiotics is, and the differences with probiotic.

REPLY: Thank you for the comment. Following the comments by Reviewer 4, we have now cited the definition of a prebiotic versus probiotic, as outlined by the International Scientific Association for Probiotics and Prebiotics (ISAPP) in their latest consensus statement.

Pg 2, L71: an explanation of the importance of a systematic review in this field is missing.

REPLY: Thank you for the comment. We have now added the sentence, “Despite significant public interest in pre- and probiotics, systematic reviews of the effect of prebiotics/probiotics on ASD and its associated behavioral/GI symptoms are lacking. Therefore, this review aims to examine the clinical role of prebiotics and probiotics in the management of GI and core ASD symptoms.”

Pg 2, L72: I suspect that sometimes probiotics are not called probiotics (especially when a single bacteria is used). How did the authors check for this issue?

REPLY: We have followed the definition outlined by the International Scientific Association for Probiotics and Prebiotics (ISAPP) in their latest consensus statement.

Pg 2, L75: where the PRISMA guideline can be found?

REPLY: This can be found online and was outlined in Figure 1.

Pg 2, L79: are there additional studies to be added since September? If yes, please update! Maybe these ones? https://www.ncbi.nlm.nih.gov/pubmed/30625189 and https://www.ncbi.nlm.nih.gov/pubmed/30743242

REPLY: Thank you for the comment. We have updated our search results up till 1st Feb 2019. Only one new study was included in our review. The study (https://www.ncbi.nlm.nih.gov/pubmed/30743242) was excluded from our systematic review as it did not investigate the effects of pre/probiotic supplementation.

Pg 5, L137: the authors cited the serotonin as an important pathway linking gut-brain axis and autism. This affirmation is actually based in a review, not an actual study. Moreover, there are other links, perhaps even stronger, that could explain the role of pre/probiotics and autism (eg. oxytocin).

REPLY: Thank you for the comment. We have now cited additional references highlighting the link between oxytocin and the pathogenesis of autism.

P6, L162: authors mentioned that there are several preclinical and clinical studies with pre-probiotic and autism but cite just one manuscript. Are there others (yes!) or should you rephrase the sentence?

REPLY: Additional studies are now cited.

P6, L190: how metabolomics can help to understand the mechanisms of pre-probiotics?

REPLY: We have now added the statement, “For future work, recent technological advancements in the area of metabolomics have vastly improved the sensitivity and accuracy at which metabolites can be detected and characterized [40].”

Reviewer 2 Report

1. Interesting review, however the literature is limited to make conclusions.

2. Prebiotics should be removed from the title and the whole manuscript since only 1 work is reviewed. Also there is no discussion by the authors as was made with probiotics. It would be better to focus only in probiotics.

3. Lines 103-116. This is the only part where the studies with probiotics and prebiotics are discussed. The rest discussion part is about the proposed action of probiotics only NOT prebiotics. This is why I insist to delete prebiotics.

4. The manuscript should be totally changed since no clinical studies are available and limited to 6. It is better to focus on the proposed action of probiotic in children with ASD and just to refer these 6 clinical studies. The methods should be deleted and the manuscript should be transformed to usual review manuscript.

5. Lines 31-33. This is not the conclusion of the study and should be removed. The lack of studies does not mean that there is no effect. The same applies for lines 180-181.

Author Response

1. Interesting review, however the literature is limited to make conclusions.

REPLY: Thank you for the comment. We agree with the reviewer, and in our manuscript, we acknowledge that “there is a general paucity of randomized, controlled trials examining the use of prebiotics or probiotics in children with ASD.”

2. Prebiotics should be removed from the title and the whole manuscript since only 1 work is reviewed. Also there is no discussion by the authors as was made with probiotics. It would be better to focus only in probiotics.

REPLY: Thank you for the comment. In our manuscript, we acknowledge that “there is a general paucity of randomized, controlled trials examining the use of prebiotics or probiotics in children with ASD.” We have decided to study both pre- and probiotics despite the apparent lack of studies as these are of significant public and research interest. After updating our search strategy, we have found 1 new study conducted on prebiotics as well.

3. Lines 103-116. This is the only part where the studies with probiotics and prebiotics are discussed. The rest discussion part is about the proposed action of probiotics only NOT prebiotics. This is why I insist to delete prebiotics.

REPLY: We have decided to study both pre- and probiotics despite the apparent lack of studies as these are of significant public and research interest. We also believe that these are both very relevant for gut microbiota and gut health.

4. The manuscript should be totally changed since no clinical studies are available and limited to 6. It is better to focus on the proposed action of probiotic in children with ASD and just to refer these 6 clinical studies. The methods should be deleted and the manuscript should be transformed to usual review manuscript.

REPLY: We have followed the usual structure for a review manuscript, which includes a methods section.

5. Lines 31-33. This is not the conclusion of the study and should be removed. The lack of studies does not mean that there is no effect. The same applies for lines 180-181.

REPLY: Thank you for the comment. We have now rephrased our statement to read, “Although the dearth of studies does not mean that there is no effect; only two open-label trials found significant improvement in GI symptoms after probiotic supplementation; three open-label trials found that probiotics helped to mitigate the behavioral issues characteristic of ASD.” We try to ensure that our conclusions are scientifically sound and not misleading.

Reviewer 3 Report

Dear Authors,

 After the review process, I have several comments:

- you should change ”... in the gut flora,...” with gut microbiota; 

- you should change the name of section 2 because the paper is a review not a research article; -  you should change the name of sections 3 and 4; 

- you should insert obligatory, in section 4, complementary strategy in order to modulate microbiota fingerprint (for example, the review https://doi.org/10.2174/1381612824666181001154242); 

- you should insert references for Line 161-169, page 6; 

- you should eliminate ”results” and ”methods” from the abstract section.

Best regards!

Author Response

After the review process, I have several comments:

- you should change ”... in the gut flora,...” with gut microbiota;

REPLY: Thank you for the comment. We have changed all “gut flora” to “gut microbiota”.

- you should change the name of section 2 because the paper is a review not a research article; you should change the name of sections 3 and 4;

REPLY: We have abided by the structure for review articles as outlined by the journal.

- you should insert obligatory, in section 4, complementary strategy in order to modulate microbiota fingerprint (for example, the review https://doi.org/10.2174/1381612824666181001154242);

REPLY: Thank you for the reference. The suggested reference has now been cited.

- you should insert references for Line 161-169, page 6;

REPLY: References have been inserted.

- you should eliminate ”results” and ”methods” from the abstract section.

REPLY: We have abided by the structure for review articles as outlined by the journal.

Reviewer 4 Report

Manuscript is relatively short for systematic review paper. I have some concerns and suggestion. Authors have indicated inclusion criteria is 'clinical trials'; however, studies on rats/mice are mentioned in line 155-160. I suggest authors to create separate heading as the Limitations of the study and include information from line 161-178 under this section. Under the limitation, authors can mentioned why they included studies conducted in animal model. There are some minor suggestions that need to be addressed. 

Please find the attached reviewed  document. . 

Author Response

Manuscript is relatively short for systematic review paper. I have some concerns and suggestion. Authors have indicated inclusion criteria is 'clinical trials'; however, studies on rats/mice are mentioned in line 155-160. I suggest authors to create separate heading as the Limitations of the study and include information from line 161-178 under this section. Under the limitation, authors can mentioned why they included studies conducted in animal model. There are some minor suggestions that need to be addressed.

REPLY: Thank you for the comment. We have abided by the structure for review articles as outlined by the journal.

Please find the attached reviewed document.

REPLY: Thank you for the minor changes suggested. We have made the necessary changes.

Reviewer 5 Report

Manuscript ID: medicina-460006

Title: A Systematic Review of the Role of Prebiotics and Probiotics in Autism Spectrum Disorders.

General comments:

Overall, well written and interesting publication.

The main problem is, that according to the title, the reader expects accurate discussion of the current knowledge on the subject. The title is misleading and does not match to the content – it should emphasise that this a kind of short analysis (meta-analysis?) of clinical trials.

Detailed comments:

1. Authors should use term ‘gut microbiota’ or ‘microbiota’ instead ‘gut flora’ which is incorrect, inaccurate and misleading. ‘Flora’ is associated with plants not GI tract.

2. The following references could be considered for including to the text:

Enzo Grossi, Sara Melli, Delia Dunca et al.  Unexpected improvement in core autism spectrum disorder symptoms after long-term treatment with probiotics, 2016 https://doi.org/10.1177/2050313X16666231

Zhai Q, Cen S, Jiang J, Zhao J, Zhang H, Chen W. Disturbance of trace element and gut microbiota profiles as indicators of autism spectrum disorder: A pilot study of Chinese children.

Environ Res. 2019, 501-509. doi: 10.1016/j.envres.2019.01.060.

Megan R. Sanctuary, Jennifer N. Kain, Shin Yu Chen et al. Pilot study of probiotic/colostrum supplementation on gut function in children with autism and gastrointestinal symptoms. PLOSone https://doi.org/10.1371/journal.pone.0210064

3. Lines 57-58: give references to the definition of ‘probiotics’. There is newer one: Hill, C., F. Guarner, G. Reid, G. R. Gibson, D. J. Merenstein, B. Pot, L. Morelli, R. B. Canani, H. J. Flint, S. Salminen, et al. 2014. The International Scientific Association for Probiotics and Prebiotics consensus statement on the scope and appropriate use of the term probiotic. Nature Reviews Gastroenterology & Hepatology 11(8):506–14.

4. Line 67: accordingly give definition of prebiotics with references.

5. In Results: Fig 1 demands description. Also Table 1 should be described in the text.

Author Response

General comments:

Overall, well written and interesting publication.

The main problem is, that according to the title, the reader expects accurate discussion of the current knowledge on the subject. The title is misleading and does not match to the content – it should emphasise that this a kind of short analysis (meta-analysis?) of clinical trials.

REPLY: Thank you for the comment. This is a systematic review in accordance with PRISMA guidelines for systematic reviews/meta-analyses. This is not a meta-analysis as no statistical analysis was feasible given the paucity and heterogeneity of studies.

Detailed comments:

1. Authors should use term ‘gut microbiota’ or ‘microbiota’ instead ‘gut flora’ which is incorrect, inaccurate and misleading. ‘Flora’ is associated with plants not GI tract.

REPLY: Thank you for the comment. We have changed all “gut flora” to “gut microbiota”.

2. The following references could be considered for including to the text:

Enzo Grossi, Sara Melli, Delia Dunca et al.  Unexpected improvement in core autism spectrum disorder symptoms after long-term treatment with probiotics, 2016 https://doi.org/10.1177/2050313X16666231

Zhai Q, Cen S, Jiang J, Zhao J, Zhang H, Chen W. Disturbance of trace element and gut microbiota profiles as indicators of autism spectrum disorder: A pilot study of Chinese children. Environ Res. 2019, 501-509. doi: 10.1016/j.envres.2019.01.060.

Megan R. Sanctuary, Jennifer N. Kain, Shin Yu Chen et al. Pilot study of probiotic/colostrum supplementation on gut function in children with autism and gastrointestinal symptoms. PLOSone https://doi.org/10.1371/journal.pone.0210064

REPLY: Thank you for the relevant references. We have included them in our manuscript.

3. Lines 57-58: give references to the definition of ‘probiotics’. There is newer one: Hill, C., F. Guarner, G. Reid, G. R. Gibson, D. J. Merenstein, B. Pot, L. Morelli, R. B. Canani, H. J. Flint, S. Salminen, et al. 2014. The International Scientific Association for Probiotics and Prebiotics consensus statement on the scope and appropriate use of the term probiotic. Nature Reviews Gastroenterology & Hepatology 11(8):506–14.

REPLY: Thank you for the reference. We have updated our definition for prebiotics and probiotics.

4. Line 67: accordingly give definition of prebiotics with references.

REPLY: Thank you for the comment. We have updated our definition for prebiotics and probiotics.

5. In Results: Fig 1 demands description. Also Table 1 should be described in the text.

REPLY: Thank you for the comment. We have added further description and elaboration to Figure 1 and Table 1.

“A total of seven clinical trials were systematically reviewed (Table 1). One clinical trial examined the use of prebiotic and/or diet exclusion while six involved the use of probiotic supplementation in children with ASD. Most of these were prospective, open-label studies, with the exception of three randomized, double-blind, placebo-controlled trials.”

Round 2

Reviewer 2 Report

The manuscript has been improved.

Please correct the format of table 1.

Reviewer 3 Report

Dear Authors,

I do not have any supplementary comments.

Best regards!